# Comparison of Long-Term Oncological Results in Young Women with Breast Cancer between BRCA-Mutation Carriers Versus Non-Carriers: How Tumor and Genetic Risk Factors Influence the Clinical Prognosis

**DOI:** 10.3390/cancers15164177

**Published:** 2023-08-19

**Authors:** Corrado Tinterri, Simone Di Maria Grimaldi, Andrea Sagona, Erika Barbieri, Shadya Darwish, Alberto Bottini, Giuseppe Canavese, Damiano Gentile

**Affiliations:** 1Breast Unit, IRCCS Humanitas Research Hospital, Via Manzoni 56, 20089 Rozzano, Milan, Italy; corrado.tinterri@hunimed.eu (C.T.); simone.dimariagrimaldi@cancercenter.humanitas.it (S.D.M.G.); andrea.sagona@cancercenter.humanitas.it (A.S.); erika.barbieri@cancercenter.humanitas.it (E.B.); shadya.darwish@humanitas.it (S.D.); alberto.bottini@cancercenter.humanitas.it (A.B.); giuseppe.canavese@cancercenter.humanitas.it (G.C.); 2Department of Biomedical Sciences, Humanitas University, Via Rita Levi Montalcini 4, 20090 Pieve Emanuele, Milan, Italy

**Keywords:** breast cancer, young women, BRCA, surgery, neoadjuvant chemotherapy, cancer susceptibility genes, breast cancer prognosis

## Abstract

**Simple Summary:**

Breast cancer (BC) is still the most prevalent malignancy diagnosed in young women (YW) (aged 18–40 years). Additionally, BC is considered the leading cause of cancer-related deaths in YW. A younger age is also associated with a higher risk of harboring a BRCA mutation. Previous studies investigating the impact of BRCA mutation on clinical prognosis reported conflicting results. Until today, it is unclear whether a germline BRCA mutation has independent prognostic implications after an initial BC diagnosis. To further investigate the influence of BRCA mutation on the clinical outcomes of young BC patients, we performed a retrospective analysis with the primary aim of evaluating the characteristics of YW with BC, comparing the long-term oncological results between BRCA-mutation carriers and non-carriers.

**Abstract:**

Background: Breast cancer (BC) is very uncommon in young women (YW) and it is unclear whether a BRCA mutation has prognostic implications. Our aim was to evaluate the characteristics of YW with BC by comparing the long-term oncological results between BRCA-mutation carriers and non-carriers. Methods: We retrospectively reviewed all the consecutive YW (aged 18–40 years) diagnosed with BC. Endpoints were disease-free survival (DFS), distant disease-free survival (DDFS), and overall survival (OS). Results: 63 YW with a BRCA mutation were compared with 339 YW without BRCA mutation. BRCA-mutation carriers were younger (60.3% versus 34.8% if age ≤ 35 years, *p* = 0.001) and presented with more aggressive tumors (66.7% versus 40.7% if G3, *p* = 0.001; 57.2% versus 12.4% if biological subtype triple-negative, *p* = 0.001; 73.0% versus 39.2% if Ki67 ≥ 25%, *p* = 0.001). Non-carriers presented significantly better DFS, DDFS, and OS compared with BRCA-mutation carriers. Neoadjuvant chemotherapy was found to be an independent protective factor for OS in BRCA-mutation carriers. Conclusions: BC is more likely to present at a younger age (≤ 35 years) and with more aggressive characteristics (G3, triple-negative, Ki67 ≥ 25%) in YW with BRCA mutation compared with their non-mutated counterparts. Young BRCA-mutation carriers showed a poorer prognosis in terms of recurrence and survival compared with non-carriers. The implementation of neoadjuvant chemotherapy may improve survival in YW with BC and BRCA mutation.

## 1. Introduction

Breast cancer (BC) is very uncommon in young adults, yet it is still the most prevalent malignancy diagnosed in women 40 years of age and younger [1]. Up to 10% of women diagnosed with BC are younger than 40, with an estimated 12,000 cases diagnosed in this age group annually in the United States [2]. Despite the fact that the incidence of BC is age-dependent, a constant increase in BC diagnoses in young women (YW) has been recently reported in many countries [3,4,5,6]. This sudden rise in BC cases in YW is of crucial importance since the behavior of these tumors is, in the majority of cases, more aggressive than those that develop in older women [7,8,9]. Additionally, BC is considered the leading cause of cancer-related deaths in women under the age of 40 [10,11]. The reasons why YW with BC experience a worse prognosis are complex and depend on several factors. Firstly, YW are more likely to present with symptoms and at a more advanced stage, compared with older women, in part due to diagnostic delays and lack of accurate screening [12,13]. Secondly, BC in younger patients typically presents with more unfavorable pathologic characteristics (i.e., high histological grade, lymphovascular invasion), a higher rate of local recurrences [14,15], and more aggressive subtypes, such as HER2-positive and triple-negative tumors [16,17]. Lastly, younger age is associated with a higher risk of harboring a BC-predisposing gene mutation, with a significantly higher probability of detecting a BRCA mutation in women younger than 35 years of age than in the general population (9.4% versus 0.2%, respectively) [18,19]. Clinicopathological characteristics that distinguish BRCA-associated BC from sporadic cancer include a higher histological grade, hormone receptor negativity, early-onset of disease, and an elevated risk of synchronous bilateral BC [20,21,22,23,24]. Even though BRCA-mutation carriers only have a 10% chance of developing BC, knowledge of BRCA mutational status is of crucial importance for treatment planning (including bilateral prophylactic mastectomy and salpingo-oophorectomy) [25,26]. Previous retrospective analyses, prospective cohort studies, and meta-analyses investigated the impact of BRCA mutation on the prognosis of YW with BC and compared it with patients with sporadic BC, reporting conflicting findings [27,28,29,30,31,32,33]. Until today, it was unclear whether a germline BRCA mutation had independent prognostic implications after an initial BC diagnosis. To further investigate the influence of BRCA mutation on the clinical outcomes of young BC patients, we performed a retrospective analysis with the primary aim of evaluating the characteristics of YW with BC comparing the long-term oncological results between BRCA-mutation carriers and non-carriers.

## 2. Materials and Methods

### 2.1. Study Design and Management of BRCA-Mutation Carriers

We retrospectively reviewed all the consecutive YW (aged 18–40 years) diagnosed with primary BC treated at the Breast Unit of the IRCCS Humanitas Research Hospital (Milan, Italy), between February 2008 and March 2019. All histological subtypes were included. All YW underwent mutation analysis at the same institution. Genetic testing for BRCA1 and BRCA2 pathogenic variants was performed in BC subjects meeting specific characteristics, such as age ≤ 40 years, triple-negative tumors and age ≤ 60 years, ovarian cancer or bilateral BC and age ≤ 50 years, synchronous or metachronous ovarian and BC regardless of age. Familial predisposition factors were also taken into consideration, offering genetic testing to BC patients < 50 years of age with a first-degree relative diagnosed with BC at < 50 years of age, a first-degree relative with ovarian cancer, or a first-degree relative with bilateral BC. Additionally, genetic testing was made available to BC patients diagnosed at any age with two first-degree relatives with BC or ovarian cancer. Genetic testing for BRCA1 and BRCA2 pathogenic variants could be carried out either before or after surgery. If the mutation analysis had been performed prior to the operation, surgical treatment of the affected and contralateral breast (breast-conserving surgery versus unilateral or bilateral mastectomy) would have been managed in accordance with patient preferences [26]. Following surgery, each patient’s adjuvant therapies were discussed by a multidisciplinary tumor board composed of breast surgeons, breast medical oncologists, radiotherapists, radiologists, gynecologists, plastic surgeons, geneticists, and pathologists. A password-protected institutional database that was made ad hoc was used to prospectively compile the demographic, clinical, tumor, and pathologic characteristics of YW with BC. Details of pre-operative treatment, type of surgery, and adjuvant therapies were retrospectively evaluated and analyzed. For further analyses, YW with BC were divided into two groups: BRCA-mutation carriers versus non-carriers. BRCA1 and BRCA2-mutation carriers were grouped together. Patient, tumor, surgical treatment, and post-operative data were compared between the two groups. The following exclusion criteria were applied: patients aged <18 or >40 years, BRCA-mutation carriers without BC or with ovarian cancer as the first presentation, follow-up <50 months, lost to follow-up. Each patient gave informed consent for surgery and clinical data collection.

### 2.2. Endpoints and Definitions

The primary endpoint of the study was to evaluate YW with BC and compare the long-term oncological results between BRCA-mutation carriers and non-carriers in terms of disease-free survival (DFS), distant disease-free survival (DDFS), and overall survival (OS). DFS was defined as the period from the date of surgical treatment for BC to the date of any tumor progression including loco-regional recurrence or distant metastasis. DDFS was defined as the period from the date of surgery for BC and the date of detection of distant metastasis. OS was defined as the time interval from BC treatment to death from any cause or to the date of last contact. Estrogen receptor and progesterone receptor expression levels were assessed by standard immunohistochemical techniques. HER2 status was assessed by immunohistochemistry and defined as negative if the score was 0/1+, equivocal if the score was 2+, or positive if the score was 3+. Equivocal cases were further assessed by fluorescent in situ hybridization, according to the recommendations of the American Society of Clinical Oncology/College of American Pathologists (ASCO/CAP) [34]. Tumor molecular subtype was defined according to the St. Gallen 2013 classification [35].

### 2.3. Statistical Analysis

The last follow-up of YW with BC was updated up to 24 April 2023. Patient and tumor characteristics were presented according to mutational status (BRCA-mutation carriers versus non-carriers) and reported as median and range for continuous variables and frequencies (No., %) for categorical variables. Differences in demographic, clinicopathological, and treatment characteristics between the two groups (BRCA-mutation carriers versus non-carriers) were compared using the chi-square test or Fisher’s exact test. For recurrence and survival analyses, patients were divided into different groups based on their mutational status. The Kaplan–Meier method was used to generate the recurrence and survival curves and to estimate the DFS, DDFS, and OS rates. Multivariate analyses were performed using the Cox proportional hazards model to identify independent risk and protective factors of DFS, DDFS, and OS. Hazard ratios and 95% confidence intervals were calculated. Statistical significance was set at *p* < 0.05; all statistical tests were two-tailed. Data analyses and figures were performed with the IBM SPSS 25.0 software.

## 3. Results

### 3.1. Comparison of Characteristics in Young Women with Breast Cancer

A total of 402 BC patients aged 18–40 years were included in the study. All YW with BC underwent genetic testing and surgery at the Breast Unit of IRCCS Humanitas Research Hospital (Milan, Italy). Of these, 63 patients tested positive for inherited BRCA1 and BRCA2 pathogenic variants. The majority of these patients (82.5%) underwent genetic testing post-operatively. The characteristics of the 63 YW with BRCA mutation were compared with 339 YW without BRCA mutation. The median age was 35 years (range, 24–40) in BRCA-mutation carriers and 37 years (range, 22–40) in non-carriers. The median diameter of the breast tumor was 21 mm (range, 5–100) in BRCA-mutation carriers and 18 mm (range, 4–110) in non-carriers. The majority of BRCA-mutation carriers presented with G3 (66.7%) and triple-negative tumors (57.2%). The majority of non-carriers presented with pT1 (56.6%) and luminal-like tumors (64.3%). Overall, 43 YW (10.7%) underwent neoadjuvant chemotherapy (BRCA-mutation carriers 11.1% versus non-carriers 10.6%, *p* = 0.908). In terms of surgical treatment, the majority of BRCA-mutation carriers underwent mastectomy (52.4%) and axillary lymph node dissection (50.8%); however, no statistically significant difference was observed between the two groups (*p* = 0.058 and *p* = 0.163, respectively). A higher proportion of non-carriers were treated with adjuvant endocrine therapy (BRCA-mutation carriers 42.9% versus non-carriers 81.1%, *p* < 0.001). Demographic, tumor characteristics, and treatments received according to BRCA mutational status were detailed in Table 1.

At multivariate analysis, numerous statistically significant differences in terms of demographic and tumor characteristics were found between the two groups. BRCA-mutation carriers were younger (60.3% versus 34.8% if age ≤ 35 years, odds ratio [OR] = 17.699, 95% confidence interval [95%CI] = 33.871–35.568, *p* = 0.001) and presented with more aggressive tumors (66.7% versus 40.7% if G3, OR = 17.119, 95%CI = 2.549–2.828, *p* = 0.001; 57.2% versus 12.4% if biological subtype triple-negative, OR = 52.727, 95%CI = 2.042–2.417, *p* = 0.001; 73.0% versus 39.2% if Ki67 ≥ 25%, OR = 58.981, 95%CI = 47.135–58.505, *p* = 0.001). Univariate and multivariate analyses were summarized in Table 1.

### 3.2. Long-Term Oncological Results and Independent Predictive Factors for Clinical Prognosis

At a median follow-up of 105 months (range, 50–170), 54 YW with BC (/402, 13.4%) experienced recurrence. In the BRCA-mutation carriers group, nine patients (/63, 14.3%) had a loco-regional BC recurrence, eleven patients (/63, 17.5%) had a contralateral BC recurrence, and twelve patients (/63, 19.1%) had a distant recurrence. In the same group, five patients experienced a second primary malignancy (two patients had ovarian cancer, one patient had melanoma, and two patients developed acute myeloid leukemia). In the non-carriers group, 13 patients (/339, 3.8%) had a loco-regional BC recurrence, 18 patients (/339, 5.3%) had a distant recurrence. Non-carriers presented with significantly better long-term oncological results in terms of DFS and DDFS compared with BRCA-mutation carriers. The DFS rate at 3, 5, and 10 years was 85.7%, 80.9%, 58.1%, and 96.8%, 95.9%, 91.1%, in BRCA-mutation carriers versus non-carriers (*p* < 0.001), respectively. The DDFS rate at 3, 5, and 10 years was 93.7%, 87.1%, 76.1%, and 99.1%, 97.3%, 91.2%, in BRCA-mutation carriers versus non-carriers (*p* = 0.003), respectively. Figure 1 and Figure 2 depict the comparison of the recurrence curves between the two groups.

Overall, twelve YW with BC died; six BRCA-mutation carriers (/63, 9.5%) and six non-carriers (/339, 1.8%). Non-carriers presented with significantly better long-term oncological results in terms of OS compared with BRCA-mutation carriers. The OS rate at 3, 5, and 10 years was 98.4%, 95.1%, 87.8%, and 100%, 99.7%, 98.0%, in BRCA-mutation carriers versus non-carriers (*p* = 0.002), respectively. Figure 3 depicts the comparison of the survival curves between the two groups.

Neoadjuvant chemotherapy was found to be an independent protective factor for OS in YW with BRCA mutation (hazard radio [HR] = 14.885, 95%CI = 2.343–94.566, *p* = 0.004). Independent predictive factors for clinical prognosis were summarized in Table 2.

## 4. Discussion

We performed a retrospective analysis with a long follow-up period, showing a significant difference in terms of DFS, DDFS, and OS between YW carrying a BRCA mutation and YW without this mutation. These discrepancies in oncological outcomes can be partly explained by differences in the pathogenic features and patterns of BC subtypes of tumors arising in the two different groups of YW.

Important issues about the pathology and biology of BC in YW have been investigated by Azim et al. [36], who performed a pooled gene expression analysis on two independent cohorts of patients and evaluated the association between the patients’ age and nearly 50 genes related to early-onset of disease. The analysis was adjusted for differences in BC molecular subtype, dimension, tumor grade, and nodal status. The results showed that, independent of subtype, stage, and grade, YW show higher expression of c-kit, RANK-ligand, mammary stem cells, luminal progenitors, and BRCA1-mutation signatures. The high BRCA1-mutation signature expression is consistent with the known high prevalence of BRCA1 mutation in YW with BC [37], and these patients are commonly diagnosed with basal-like tumors [38]. Similarly, Huzarski et al. [32] performed a large retrospective analysis including 3345 patients; of whom 233 (7.0%) carried a BRCA1 mutation. BRCA1-mutation carriers were significantly younger (mean age 41.9 years versus 44.1 years, *p* < 0.001), and presented with more triple-negative tumors (84.1% versus 38.1%, *p* < 0.001). Additionally, BRCA1 status was associated with a (nonsignificant) worse prognosis than non-carriers; however, after adjusting for other prognostic features, there was a significant difference in mortality between carriers and non-carriers (HR = 1.81, 95%CI = 1.26–2.61, *p* = 0.002). Lymph node status was highly predictive of survival among BRCA1-mutation carriers.

The presence of more aggressive tumor features, including a high tumor grade, triple-negative tumors, and Ki67 ≥ 25%, may explain why YW with BRCA mutation showed a worse clinical prognosis in terms of recurrence and survival compared with non-carriers; however, previous studies reported conflicting results.

The POSH trial [33] was a prospective multicenter study of YW (aged 18–40 years) at first diagnosis of invasive BC, aiming at determining the effect of germline BRCA mutation on clinical outcomes. The study recruited 2733 YW; of these, 338 patients (12.4%) had a pathogenic BRCA mutation. There was no significant difference in OS between BRCA-mutation carriers and non-carriers at any timepoint; in fact, the OS rate at 2, 5, and 10 years was 97.0%, 83.8%, 73.4%, and 96.6%, 85.0%, 70.1%, in BRCA-mutation carriers versus non-carriers (*p* = 0.76), respectively. Rennert G et al. [30] performed a national population-based retrospective study of Israeli women to determine the influence of BRCA mutation on the clinical prognosis. A BRCA mutation was identified in 10% of the women who were of Ashkenazi Jewish background. The adjusted HRs for death from BC were not significantly different among BRCA-mutation carriers and non-carriers. Moreover, a statistically significant interaction between BRCA1 mutation status and chemotherapy was found for OS (*p* = 0.002). Verhoog et al. [31] performed a small cohort retrospective analysis, comparing the oncological outcomes of 49 BRCA-mutation carriers with those of 196 non-carriers. The DFS rate at 5 years was 49% and 51%, in BRCA-mutation carriers versus non-carriers (*p* = 0.98), respectively. The OS rate at 5 years was 63% and 69%, in BRCA-mutation carriers versus non-carriers (*p* = 0.88), respectively. Patients with BRCA-associated BC had twice as many triple-negative tumors (*p* < 0.005), and development of contralateral BC was four to five times as frequent as in the non-carriers group (*p* < 0.001).

On the other hand, Stoppa-Lyonnet et al. [39] performed a cohort study, comparing 40 patients with BRCA-associated BC with 143 patients with sporadic BC, showing that BRCA-mutation carriers presented with larger tumors (*p* = 0.03), had a higher rate of G3 tumors (*p* = 0.002), and had a higher frequency of triple-negative tumors (*p* = 0.003). At a median follow-up of 58 months, the clinical outcomes of BRCA-mutation carriers were significantly worse compared to non-carriers in terms of OS and metastasis-free interval (49% versus 85% and 18% versus 84%, respectively). Robson ME et al. [40] performed a combined analysis of two retrospective cohorts of Ashkenazi Jewish women, showing that BC-specific survival was worse in women with BRCA1 mutations than in those without (the OS rate at 10 years was 62% versus 85%, *p* < 0.001, respectively). Moller P et al. [41] retrospectively analyzed the oncological outcomes of 442 patients according to their mutational status (BRCA1-mutated, BRCA2-mutated, or mutation-negative); showing that the OS rate at 5 years was 73%, 96%, and 92% in BRCA1-mutated, BRCA2-mutated, and mutation-negative patients (*p* < 0.001), respectively. Wang YA et al. [42] detected a 13.5% carrier rate of pathogenic germline mutations in 480 ethnic Chinese individuals in Taiwan, with BRCA-mutation carriers presenting worse BC specific outcomes (the DFS rate at 5 years was 73.3% versus 91.1%, *p* = 0.013, respectively). Schmidt MK et al. [29] performed a large retrospective study, investigating the clinical prognosis and the long-term survival of 6478 YW with BC and comparing BRCA-mutation carriers with non-carriers. BRCA-mutation carriers had a worse OS independent of clinicopathological and treatment characteristics compared with non-carriers (HR = 1.20, 95%CI = 0.97–1.47). Baretta Z et al. [28] performed a systematic review and meta-analysis on 60 studies, analyzing the effects of BRCA germline mutation on multiple survival outcomes of BC patients. This meta-analysis involved 105,220 BC patients; of these, 3588 (3.4%) were BRCA-mutation carriers. BRCA1-mutation carriers had worse OS (HR = 1.30, 95%CI = 1.11–1.52) and worse BC-specific survival (HR = 1.45, 95%CI = 1.01–2.07) than non-carriers. BRCA2-mutation carriers had worse BC-specific survival (HR = 1.29, 95%CI = 1.03–1.62) than non-carriers, although they had similar OS.

In our study, neoadjuvant chemotherapy was found to be an independent protective factor for survival in BRCA-mutation carriers. The use of pre-operative systemic therapies is a well-known therapeutic strategy for the management of locally-advanced BC, especially for more aggressive subtypes such as triple-negative tumors; however, its efficacy in patients with a BRCA germline mutation remains inconclusive [43]. Recently, the pre-operative use of the new PARP inhibitor Talazoparib has demonstrated efficacy and promising results in the setting of BRCA-positive operable BC [44,45]. 

It is necessary to underline that our study is a single-center analysis subject to limitations due to its retrospective design. Moreover, the vast majority of BRCA-mutation carriers underwent genetic testing post-operatively and the prognostic difference between BRCA1 and BRCA2 mutations was not investigated in this study. Additionally, racial and ethnic differences among groups were not evaluated in the present analysis. However, this study also presents some strong points. All YW with BC underwent genetic testing and were observed for a long follow-up period. Moreover, no patient was lost to follow-up.

## 5. Conclusions

In conclusion, BC is more likely to present at a younger age (≤ 35 years) and with more aggressive characteristics (G3, triple-negative, Ki67 ≥ 25%) in YW with BRCA mutation compared with their non-mutated counterparts. Young BRCA-mutation carriers showed poorer prognosis in terms of recurrence and survival compared with non-carriers. The implementation of neoadjuvant chemotherapy regimens may improve survival in YW with BC and BRCA mutation.

## Figures and Tables

**Figure 1 cancers-15-04177-f001:**
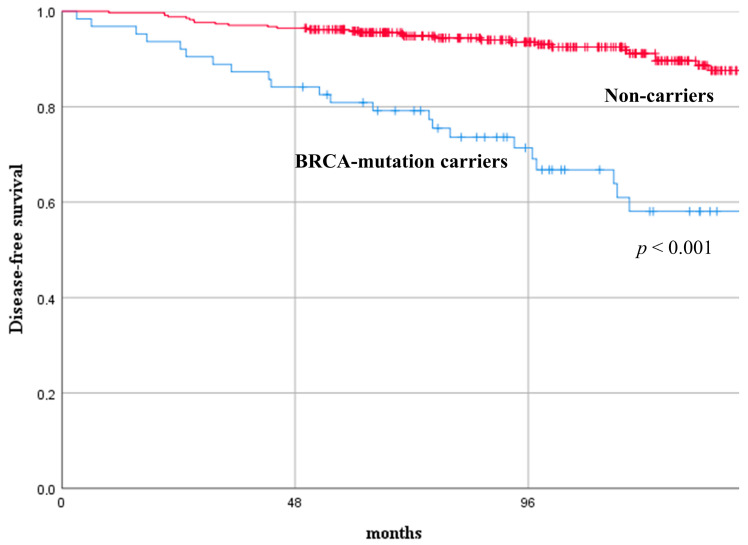
Disease-free survival curves of young women with breast cancer according to their mutational status (BRCA-mutation carriers versus non-carriers).

**Figure 2 cancers-15-04177-f002:**
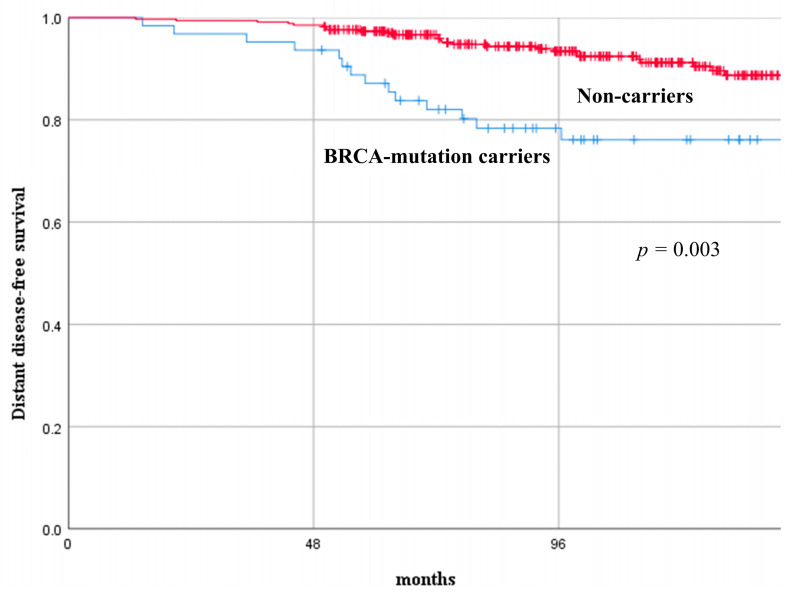
Distant disease-free survival curves of young women with breast cancer according to their mutational status (BRCA-mutation carriers versus non-carriers).

**Figure 3 cancers-15-04177-f003:**
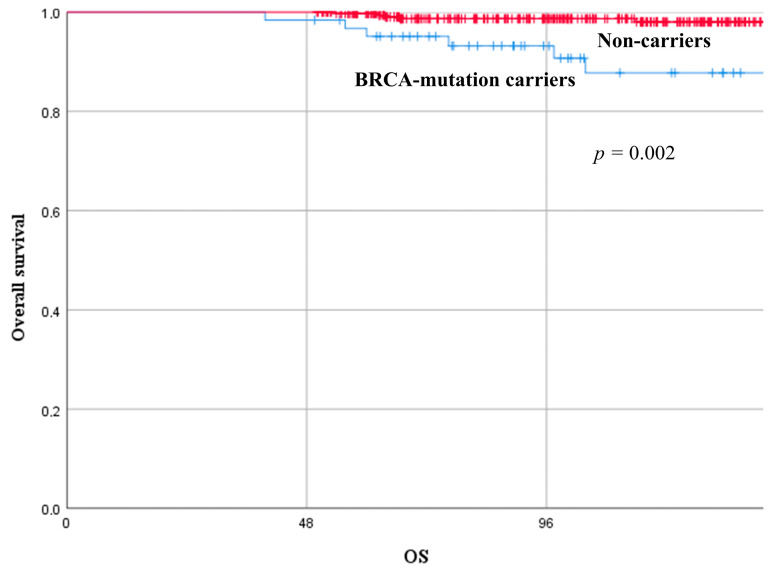
Overall survival curves of young women with breast cancer according to their mutational status (BRCA-mutation carriers versus non-carriers).

**Table 1 cancers-15-04177-t001:** Comparison of characteristics between BRCA-mutation carriers versus non-carriers.

Characteristics	BRCA-Carriers (No. 63) Tot. (%)/Median (Range)	Non-Carriers (No. 339) Tot. (%)/Median (Range)	Univariate Analysis *p*-Value	Multivariate Analysis *p*-Value OR (95%CI)
Age (years)	35 (24–40)	37 (22–40)		
- ≤35	38 (60.3%)	118 (34.8%)	<0.0001 ^a^	0.001 ^a^ 7.699 (33.871–35.568)
- >35	25 (39.7%)	221 (65.2%)	-	-
Tumor				
Grading				
- 1	0 (0%)	15 (4.4%)	<0.0001 ^a^	0.001 ^a^ 17.119 (2.549–2.828)
- 2	21 (33.3%)	186 (54.9%)	-	-
- 3	42 (66.7%)	138 (40.7%)	-	
Dimension (mm)	21 (5–100)	18 (4–110)		
- <18	22 (34.9%)	159 (46.9%)	0.08	
- ≥18	41 (65.1%)	180 (53.1%)	-	
Stage				
ypT0-is	5 (7.9%)	7 (2.1%)	0.104	
pT1	22 (34.9%)	192 (56.6%)	-	
pT2	31 (49.2%)	127 (37.5%)	-	
pT3	5 (8.0%)	13 (3.8%)	-	
pN0	34 (54.0%)	197 (58.1%)	0.008 ^a^	0.051 5.952 (0.758–1.275)
pNmic	6 (9.5%)	7 (2.1%)	-	-
pN1	11 (17.5%)	87 (25.7%)	-	
pN2	6 (9.5%)	37 (10.9%)	-	
pN3	6 (9.5%)	11 (3.2%)	-	
Biological subtype				
- Luminal-like	22 (34.9%)	218 (64.3%)	<0.0001 ^a^	0.001 ^a^ 52.717 (2.042–2.417)
- HER2+	5 (7.9%)	79 (23.3%)	-	-
- Triple negative	36 (57.2%)	42 (12.4%)	-	
- Ki67 (%)	55 (5–95)	20 (1–90)		
- <25	17 (27.0%)	206 (60.8%)	<0.0001 ^a^	0.001 ^a^ 58.981 (47.135–58.505)
- ≥25	46 (73.0%)	133 (39.2%)	-	-
Vascular invasion	24 (38.1%)	148 (43.7%)	0.414	
Treatment				
- Neoadjuvant CHT	7 (11.1%)	36 (10.6%)	0.908	
- Mastectomy	33 (52.4%)	134 (39.5%)	0.058	
- ALND	32 (50.8%)	140 (41.3%)	0.163	
- Radiotherapy	32 (50.8%)	239 (70.5%)	0.003 ^a^	
- Endocrine therapy	27 (42.9%)	275 (81.1%)	<0.0001 ^a^	
- Adjuvant CHT	46 (73.0%)	190 (56.1%)	0.005 ^a^	
- Trastuzumab	5 (7.9%)	71 (20.9%)	0.015 ^a^	

Footnotes: OR: odds ratio, 95%CI: 95% confidence interval, HER2: HER2 evaluated either on immunohistochemistry or on in situ hybridization, according to the ASCO CAP guidelines, CHT: chemotherapy, ALND: axillary lymph node dissection, ^a^: Statistically significant.

**Table 2 cancers-15-04177-t002:** Multivariate analyses of risk and protective factors of long-term oncological outcomes among young women with breast cancer.

Factors	DFS HR (95%CI) *p*-Value	DDFS HR (95%CI) *p*-Value	OS HR (95%CI) *p*-Value
BRCA			
- Carriers	Reference	Reference	Reference
- Non-carriers	** 0.203 (0.104–0.394) 0.001 **	** 0.357 (0.174–0.734) 0.005 **	0.257 (0.072–1.051) 0.059
Age (years)			
- ≤35	Reference	Reference	Reference
- >35	0.770 (0.437–1.360) 0.368	0.791 (0.445–1.408) 0.426	0.425 (0.122–1.487) 0.181
G			
-1	Reference	Reference	Reference
-2	Reference	Reference	Reference
-3	1.595 (0.799–3.185) 0.186	** 2.104 (1.037–4.265) 0.039 **	2.078 (0.391–11.053) 0.391
Biological subtype			
- Luminal-like	Reference	Reference	Reference
- HER2+	Reference	Reference	Reference
- Triple-negative	0.563 (0.316–1.002) 0.051	0.583 (0.321–1.058) 0.076	** 0.238 (0.070–0.812) 0.022 **
Ki67 (%)			
- <25	Reference	Reference	Reference
- ≥25	0.678 (0.312–1.477) 0.328	0.766 (0.356–1.651) 0.497	2.234 (0.377–13.248) 0.376
Neo-adjuvant CHT			
- No	Reference	Reference	Reference
- Yes	0.678 (0.312–1.477) 0.328	1.450 (0.517–4.069) 0.480	** 14.885 (2.343–94.566) 0.004 **
Breast surgery			
- BCS	Reference	Reference	Reference
- Mastectomy	0.770 (0.359–1.651) 0.502	1.020 (0.460–2.262) 0.962	3.862 (0.745–20.011) 0.107
ALND			
- No	Reference	Reference	Reference
- Yes	0.880 (0.468–1.653) 0.691	0.674 (0.350–1.296) 0.237	0.966 (0.226–4.132) 0.962
RT			
- No	Reference	Reference	Reference
- Yes	0.916 (0.445–1.885) 0.328	1.073 (0.503–2.810) 0.567	1.527 (0.361–6.467) 0.565
Hormone therapy			
- No	Reference	Reference	Reference
- Yes	0.584 (0.219–1.554) 0.282	0.669 (0.238–1.885) 0.447	0.160 (0.016–1.561) 0.115
Adjuvant CHT			
- No	Reference	Reference	Reference
- Yes	1.424 (0.639–3.176) 0.387	1.263 (0.568–2.810) 0.568	3.186 (0.490–20.697) 0.225
Trastuzumab			
- No	Reference	Reference	n/a
- Yes	0.687 (0.257–1.853) 0.454	0.616 (0.228–1.668) 0.340	

Footnotes: DFS: disease-free survival, DDFS: distant disease-free survival, OS: overall survival, HR: hazard ratio, 95%CI: 95% confidence interval, HER2: HER2 evaluated either on immunohistochemistry or on in situ hybridization, according to the ASCO CAP guidelines, CHT: chemotherapy, BCS: breast-conserving surgery, ALND: axillary lymph node dissection, RT: radiotherapy, **Colored bold**: statistically significant.

## Data Availability

The data presented in this study are available in this article (and Appendix A).

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
