# Peer review of "Comparison of Long-Term Oncological Results in Young Women with Breast Cancer between BRCA-Mutation Carriers Versus Non-Carriers: How Tumor and Genetic Risk Factors Influence the Clinical Prognosis"

_cancers, 2023, doi:10.3390/cancers15164177_

Round 1
Reviewer 1 Report
The authors of the article assessed the characteristics of young women with BC by comparing long-term cancer outcomes between BRCA mutation carriers and non-carriers. It was shown that BRCA mutation carriers were younger and had more aggressive tumors. The authors showed that neoadjuvant chemotherapy is an independent protective factor in BRCA mutation carriers.
1. Why is 35 chosen as the cutoff point for age? According to WHO criteria, young people are up to 45. Is there another justification here?
2. It was not taken into account which mutation was present in BRCA1 and BRCA2?
Author Response
POINT-BY-POINT REPLY TO REVIEWERS’ COMMENTS
We thank the editors and reviewers of CANCERS for the opportunity to reply to reviewers’ comments. We do believe that now the manuscript is much more precise and will be of interest to the readers of CANCERS. Please find our replies in this document in bold black and in the manuscript in bold red.
Reviewers' Comments & Replies:
Reviewer #1: The authors of the article assessed the characteristics of young women with BC by comparing long-term cancer outcomes between BRCA mutation carriers and non-carriers. It was shown that BRCA mutation carriers were younger and had more aggressive tumors. The authors showed that neoadjuvant chemotherapy is an independent protective factor in BRCA mutation carriers.
- Why is 35 chosen as the cutoff point for age? According to WHO criteria, young people are up to 45. Is there another justification here?
Reply: We thank the reviewer for the comment.
35 years was chosen as the cutoff point for age because it was the median age of the BRCA-mutation carriers group and also because two studies [18,19] cited in the Introduction pointed out this cutoff point of age as a significantly predisposing factor of harboring a BRCA-mutation.
- It was not taken into account which mutation was present in BRCA1 and BRCA2?
Reply: We thank the reviewer for the comment.
Type of mutation of BRCA-mutation carriers is now detailed in supplementary material.

Reviewer 2 Report
Tinterri et al. manuscript provides interesting insights on the hot topic regarding BRCA mutations as independent prognostic factors for young women with breast cancer. As also highlighted by the authors in the text, previous studies reported conflicting results, making the manuscript’s topic original, timely and of high interest for the oncologic research field. The simple summary and the abstract provide a clear and focused recap of the authors’ work. The introduction part is correctly focused and does not lack essential and important information required for understanding the authors’ work. Methods section is clear and thoroughly described. The experimental setting appears to be correctly framed to support the authors’ objective. Results are clearly presented and no logical gaps between the parts are present. The discussion and the conclusion parts are correctly and fully supported by results and the state-of-the-art literature. The strength of the study is that it clearly establishes the prognostic value of BRCA mutations in young women with breast cancer, providing novel insights for a research topic with conflicting findings. However, to ensure publication suitability, this reviewer raises these minor issues as detailed in the checklist below:
· Considering the authors’ results, chosen keywords could be improved, e.g. by adding “neoadjuvant chemotherapy”, “Cancer susceptibility genes” and “Breast cancer prognosis” which fully describe and clearly fit within the conclusions of the manuscript;
· Y axis description in Figure 1 and Figure 2 is misleading, since it appears in contrast with what has been stated in the text. In this regard, with the term “recurrence probability”, one could argue that BRCA-mutations carriers have a lower tumor recurrence probability compared to non-carriers. Could “Disease-Free Survival” and “Distant Disease-Free Survival” be better used as Y axis denominators to ensure message comprehension?
· In Table 2 a clear, visual distinction (e.g. by color?) should be operated to better convey the table content message;
· The authors failed to cite the study from Wang et al (Germline breast cancer susceptibility gene mutations and breast cancer outcomes. BMC Cancer. 2018 Mar 22;18(1):315. doi: 10.1186/s12885-018-4229-5) which supports the authors’ results and conclusions.
· Although they are strictly supportive data, supplementary information should be in English
· The authors are invited to better detail the limitations of their study (e.g. was race considered in their retrospective study?) and to define possible further goals (e.g. among BRCA-mutation carriers, were other genes found to be relevant and significant descriptors of the different molecular characteristics of the tumors analyzed? Where any differences between BRCA1 and BRCA2 mutations found to be significant for their prognostic value?)
The authors should submit their manuscript for a deep proof-reading from a native speaker to ensure the correct use of the English language.
Author Response
POINT-BY-POINT REPLY TO REVIEWERS’ COMMENTS
We thank the editors and reviewers of CANCERS for the opportunity to reply to reviewers’ comments. We do believe that now the manuscript is much more precise and will be of interest to the readers of CANCERS. Please find our replies in this document in bold black and in the manuscript in bold red.
Reviewers' Comments & Replies:
Reviewer #2: Tinterri et al. manuscript provides interesting insights on the hot topic regarding BRCA mutations as independent prognostic factors for young women with breast cancer. As also highlighted by the authors in the text, previous studies reported conflicting results, making the manuscript’s topic original, timely and of high interest for the oncologic research field. The simple summary and the abstract provide a clear and focused recap of the authors’ work. The introduction part is correctly focused and does not lack essential and important information required for understanding the authors’ work. Methods section is clear and thoroughly described. The experimental setting appears to be correctly framed to support the authors’ objective. Results are clearly presented and no logical gaps between the parts are present. The discussion and the conclusion parts are correctly and fully supported by results and the state-of-the-art literature. The strength of the study is that it clearly establishes the prognostic value of BRCA mutations in young women with breast cancer, providing novel insights for a research topic with conflicting findings. However, to ensure publication suitability, this reviewer raises these minor issues as detailed in the checklist below:
- Considering the authors’ results, chosen keywords could be improved, e.g. by adding “neoadjuvant chemotherapy”, “Cancer susceptibility genes” and “Breast cancer prognosis” which fully describe and clearly fit within the conclusions of the manuscript;
Reply: We thank the reviewer for the comment.
The keywords have been improved and updated according with the reviewer’s suggestions.
- Y axis description in Figure 1 and Figure 2 is misleading, since it appears in contrast with what has been stated in the text. In this regard, with the term “recurrence probability”, one could argue that BRCA-mutations carriers have a lower tumor recurrence probability compared to non-carriers. Could “Disease-Free Survival” and “Distant Disease-Free Survival” be better used as Y axis denominators to ensure message comprehension?
Reply: We thank the reviewer for the comment.
Figures have been modified according with the reviewer’s suggestions.
- In Table 2 a clear, visual distinction (e.g. by color?) should be operated to better convey the table content message;
Reply: We thank the reviewer for the comment.
In Table 2, differences between groups are now highlighted in Colored bold.
- The authors failed to cite the study from Wang et al (Germline breast cancer susceptibility gene mutations and breast cancer outcomes. BMC Cancer. 2018 Mar 22;18(1):315. doi: 10.1186/s12885-018-4229-5) which supports the authors’ results and conclusions.
Reply: We thank the reviewer for the comment.
This study was cited in the Discussion section.
- Although they are strictly supportive data, supplementary information should be in English
Reply: We thank the reviewer for the comment.
All supplementary information are now in English language.
- The authors are invited to better detail the limitations of their study (e.g. was race considered in their retrospective study?) and to define possible further goals (e.g. among BRCA-mutation carriers, were other genes found to be relevant and significant descriptors of the different molecular characteristics of the tumors analyzed? Where any differences between BRCA1 and BRCA2 mutations found to be significant for their prognostic value?)
Reply: We thank the reviewer for the comment.
Limitations were modified according with the reviewer’s suggestions.
The English language was revised by a native English speaker.
